# Tribocorrosion Properties of NiCrAlY Coating in Different Corrosive Environments

**DOI:** 10.3390/ma13081864

**Published:** 2020-04-16

**Authors:** Bo Li, Yimin Gao, Cong Li, Hongjian Guo, Qiaoling Zheng, Yefei Li, Yunchuan Kang, Siyong Zhao

**Affiliations:** 1State Key Laboratory for Mechanical Behaviour of Materials, School of Materials Science and Engineering, Xi’an Jiaotong University, Xi’an 710049, China; ymgao@xjtu.edu.cn (Y.G.); zhengql@mail.xjtu.edu.cn (Q.Z.); yefeili@126.com (Y.L.); kangyc30@stu.xjtu.edu.cn (Y.K.); 2School of Bailie Mechanical Engineering, Lanzhou City University, Lanzhou 730070, China; chinaghj2019@hotmail.com; 3Guangxi Great Wall Machineries, Hezhou 542800, China; wei9786@163.com

**Keywords:** corrosion-wear performance, dense structure, corrosion potential, corrosion rate, worn surface

## Abstract

Atmospheric plasma spraying (APS) was taken to fabricate the NiCrAlY coating. The corrosion-wear properties of NiCrAlY coating was measured respectively under deionized water, artificial seawater, NaOH solution and HCl solution. Experimental results presented that the as-sprayed NiCrAlY coating consisted of Ni_3_Al, nickel-based solid solution, NiAl and Y_2_O_3_. In deionized water, the coating with the lowest corrosion current density (*i_corr_*) of 7.865 × 10^−8^ A/cm^2^ was hard to erode. Meanwhile, it presented a lower friction coefficient and the lowest wear rate. In HCl solution, NiCrAlY coating gave the highest corrosion current density (*i_corr_*) of 3.356 × 10^−6^ A/cm^2^ and a higher wear rate of 6.36 × 10^−6^ mm^3^/Nm. Meanwhile, the emergence of Al(OH)_3_ on the coating surface could reduce the direct contact between the counter ball and sample effectively, which was conducive to the lowest friction coefficient of 0.24.

## 1. Introduction

In engineering fields, wear often occurs under different corrosive circumstances leading to the degradation rate of engineering parts [1]. For instance, some mechanical parts utilized in the marine atmosphere, pulping and mining, suffer the collaborative destruction of corrosion and wear [2,3,4,5,6]. Meanwhile, the synergism of corrosion and wear decreases the service life of the material. In the process of friction, the passive film on the worn surface could be destroyed by friction force and the new passive film is hard to form, which would make the material suffering more serious damage. Normally, the corrosion-wear material loss is greater than the sum of corrosion and wear. Therefore, it is very imperative to improve the corrosion-wear resistance property of mechanical parts in different corrosive environments. To meet this requirement, the protective coatings are applied to protect the mechanical parts without changing the external structure. MCrAlY (M = Cobalt and/or Nickel) alloys with excellent oxidation resistance, corrosion resistance and wear resistance performance have been widely used in nuclear power, automotive and marine industries acting as the protective coatings [1,7,8,9,10,11,12,13,14,15]. J. Chen et al. investigated the tribocorrosion behavior of NiCoCrAlYTa coating in corrosion. The results showed that this kind of coating presented an extremely dense structural characteristic and excellent tribological performance in NaOH and HCl solutions [1]. M. Marcu et al. studied the microstructure and oxidation resistance of as-sprayed NiCrAlY/Al_2_O_3_ coating. The results presented that the as-sprayed NiCrAlY/Al_2_O_3_ coating has the best cyclic oxidation resistance with an oxidation rate of 2.62 × 10^−12^ g^2^·cm^−4^·s^−1^ at high temperature and good adhesion during the cyclic oxidization treatment [8]. Current researches mainly focus on the oxidation resistance, corrosion, mechanical and tribological performance of the coatings [16,17,18,19,20,21,22]. These materials are also used for reciprocating parts in corrosive environments [23], so the research of the wear-corrosion resistance is still important in the process of sliding. However, few researches pay attention to the synergy of corrosion and wear [24], and its mechanism is still unclear.

In this work, the tribocorrosion properties of NiCrAlY coating were studied and the synergistic mechanisms between wear and corrosion in different corrosive environments were discussed in detail. The objective of this paper is to research how corrosive environments affect the tribological behavior of NiCrAlY coating and the interaction degree between corrosion and wear. This research would provide usable direction to the NiCrAlY coating application in corrosive environments.

## 2. Materials and Methods

### 2.1. Coating Preparation

Gas atomized spherical Ni_22_Cr_10_Al_1.0_Y (wt.%) powder (53–106 µm) was bought from Sulzer Metco (Winterthur, Switzerland). The NiCrAlY coating was prepared by atmospheric plasma spraying (APS). The Inconel 718 alloy was sand-blasted, then ultrasonically cleaned with ethanol before spraying. The coating thickness was about 300 µm. The specific spraying parameters presented were: flow rate of Ar was 40 L/min; flow rate of H_2_ was 5 L/min; spraying angle was 90°; feed rate of the powder was 42 g/min; voltage was 60 V; the current was 500 A and spray distance was 110 mm.

### 2.2. Characterization

The micromorphologies of cross-section and worn surface of this coating were measured by field emission scanning electron microscopy (FE-SEM, Tescan Mira 3, Bron, Kohoutovice, Czech Republic). A Philips X’Pert-MRD X-ray diffractometer (XRD; Cu-K_a_ radiation, current 150 mA, potential 40 kV, Philips, Eindhoven, The Netherlands) was utilized to analyzed phase composition. The phase compositions on the worn surface were analyzed by Czemy-Tumer Labram HR800 Raman spectrometer (Horiba, Paris, France).

### 2.3. Tribocorrosion Tests

The tribocorrosion experiments were tested in deionized water (pH = 7), artificial seawater (pH = 8.2), 0.1 M NaOH solution (pH = 13) and 0.1 M HCl solution (pH = 1), with reciprocating ball-on-disk tribometer (UMT, Karlsruhe, Germany). The schematic diagram is shown in Figure 1. The polytetrafluoroethylene (PTFE) does not corrode as it is chemically inert to corrosion. So it acted as the solution cell material. The Al_2_O_3_ ceramic ball acted as the counter ball, whose diameter was 5 mm. Before the friction experiment, the surface of the coating was burnished till the roughness close to 0.5 µm. The tests were performed at the conditions below: room temperature, 5 N normal load, 0.8 mm/s sliding speed, 3.5 mm amplitude and 60 min duration. Repeated experiments were tested in every corrosive environments. The color 3D laser scanning microscope (VK-9710, Keyence, Osaka, Japan) and SEM were utilized to analyze the worn surface. The wear rate was got by W = V/LF, where W represented the wear rate (mm^3^/Nm), V represented the wear volume loss (mm^3^), L represented the sliding distance (m) and F represented the load (N).

## 3. Results and Discussions

### 3.1. Morphology and Composition of Powders and NiCrAlY Coating

Figure 2 presents the SEM micromorphology and XRD pattern of NiCrAlY powder. The spherical shape powder with a size of 53–106 µm (Figure 2a) exhibits satisfactory flowability and thus it is very beneficial to the feeding rate in the process of spraying [25]. The results of the XRD pattern show that the NiCrAlY powder composes of Ni_3_Al, NiAl and nickel-based solid solution and has high crystallinity (Figure 2b).

Figure 3 presents the SEM morphology of the cross-section and diffraction pattern of NiCrAlY coating. The coating contains some cracks and pores. Meanwhile, every phase combines well and between any two phases have no evident cracks (Figure 3a). Compared with the NiCrAlY powder (Figure 3b), a new phase of Y_2_O_3_ formed on the coating, which could obviously increase the microhardness and strength [26].

### 3.2. Electrochemical Performance of NiCrAlY Coating

Figure 4 gives the potentiodynamic polarization curves of NiCrAlY coating sliding conditions in different corrosive solutions. Key test parameters such as the corrosion potential (*E_corr_*), corrosion current density (*i_corr_*), anodic and cathodic Tafel slopes (*β_a_* and *β_c_*) are obtained from Figure 4 and shown in Table 1. The polarization resistance value (*R_p_*) is calculated by Stern–Geary equation:(1)$$R_{p}=\frac{\beta_{a}\times \beta_{c}}{2.303i_{corr}(\beta_{a}+\beta_{c})}.$$

Results indicate that the corrosion potential (*E_corr_*) of NiCrAlY coating under deionized water is the highest of −0.428 V (vs. SCE). However, the *E_corr_* of the coating in artificial seawater, HCl and NaOH shift to −0.516 V (vs. SCE), −0.559 V (vs. SCE) and −0.535 V (vs. SCE) respectively. Simultaneously, the corrosion current density (*i*_corr_) of this coating in deionized water shows the lowest of 7.865 × 10^−8^ A/cm^2^. Generally speaking, corrosion current density, whose rate is often used as corrosion rate, is a crucial reference to evaluate corrosion resistance [13,27]. Therefore, the coating under deionized water with the lowest corrosion rate is hard to corrode. The coating in HCl presenting the highest corrosion current density is very easy to be corroded. At the same time, the coating in deionized water has the highest *β_a_*, *β_c_* and *R_p_* of 0.072 V/dec, 0.049 V/dec and 1.610 × 10^5^ Ω respectively, which further illustrates that the coating in deionized water holds a good corrosion resistance.

### 3.3. Tribological Behavior of NiCrAlY Coating

Figure 5 shows the friction curves and wear rate of NiCrAlY coating in different corrosive solutions. The friction coefficient (COF) of the coating under the NaOH solution was the highest, with a value of 0.46. In artificial seawater and deionized water, it was 0.37 and 0.26, respectively. Surprisingly, the COF reduced to 0.24 and remains steady in HCl solution. Nevertheless, the NiCrAlY coating has a high wear rate (WR) of 6.36 × 10^−6^ mm^3^/Nm in the HCl solution. This phenomenon is likely to show the high corrosion rate of coating in HCl solution (Figure 4). The synergistic effect of corrosion and wear in a corrosive environment leads to the loss of large material, which usually larger than the synergistic effect of the sum of corrosion and wear [28,29]. So, the coating under the HCl solution presents a more obvious wear rate. The coating in the NaOH solution has the highest wear rate of 6.89 × 10^−6^ mm^3^/Nm. At the same time, the coating in deionized water gives the lowest WR of 2.36 × 10^−6^ mm^3^/Nm, which is caused by the lowest corrosion rate of coating in deionized water (Figure 4).

Figure 6 presents the 2D and 3D configurations of NiCrAlY coating worn surfaces in different corrosive solutions. The worn surface has the shallowest and narrowest friction trace in deionized water (Figure 6a,e). Therefore, the COF and WR are lower (Figure 5). It further illustrates that the coating in deionized water shows excellent corrosion and wear resistance. The worn surface of NiCrAlY coating in HCl corrosive solution is very rough and has serious corrosion (Figure 6c). So, the coating obtains high WR under HCl corrosive solution (Figure 5). The worn track of NiCrAlY coating in NaOH corrosive solution is the deepest and widest (Figure 6d,f). Therefore, this coating has the worst tribological performance (Figure 5).

To further research the influence of corrosive solution upon the corrosion-wear property of NiCrAlY coating, Raman analysis is tested. Figure 7 shows the Raman spectra of the worn surface of NiCrAlY coating in different corrosive solutions. The Al_2_O_3_, Cr_2_O_3_ and NiO are the main phases on the worn surface of NiCrAlY coating after sliding in deionized water, artificial seawater and NaOH solution. Nevertheless, the worn surface of NiCrAlY coating observes the new phase of Al(OH)_3_ after sliding in HCl corrosive solution [30]. The results indicate that the NiCrAlY coating has suffered serious corrosion in the HCl corrosive solution because of the existence of stronger and more numerous peaks [1]. The corrosion products are easily worn out during the friction process. So the wear rate of the coating under HCl solution is very high (Figure 5).

### 3.4. Lubrication Behavior of Al(OH)_3_ on NiCrAlY Coating in HCl Solution 

Figure 8 shows the corrosion-wear mechanisms of NiCrAlY coating in the HCl solution. The surface becomes very smooth because the corrosion-wear effect with the mix of oxides and hydroxides formed by electrochemical reactions (Figure 6 and Figure 7). In terms of the potential values, aluminum is the least noble element and the order of potentials follows Ni > Cr > Al [1]. So the aluminum element is more likely to be corroded at first. The following electrochemical reactions could explain the process of Al(OH)_3_ formation:(2)$$\mathrm{Al}\;\to \mathrm{Al}3^{+}+3e^{-}$$
(3)$$2H^{+}+2e^{-}\;\to H_{2}$$
(4)$$H_{2}O+2e^{-}\;\to H_{2}+2{\mathrm{OH}}^{-}$$
(5)$$\mathrm{Al}3^{+}+3{\mathrm{OH}}^{-}\;\to \mathrm{Al}{\left(\mathrm{OH}\right)}_{3}$$

Terryn et al. [31] illustrated that the generation of Al(OH)_3_ is related to local pH changes in the hydrogen reduction region. Hence, the Al(OH)_3_ could be formed where the hydrogen evolution occurs. Furthermore, when the local pH rises to above 9, Al^3+^ ions will react with excessive OH^−^ ions and forms aluminate anions [1]. Aluminate anions cannot maintain stable in HCl corrosive solution and will precipitate as Al(OH)_3_ (Figure 8). This reaction can be described as follows:(6)$$\mathrm{Al}3^{+}+4{\mathrm{OH}}^{-}\;\to {{\mathrm{AlO}}_{2}}^{-}+2H_{2}O$$
(7)$${{\mathrm{AlO}}_{2}}^{-}+H^{+}+H_{2}O\;\to \mathrm{Al}{\left(\mathrm{OH}\right)}_{3}$$

Thus, it inexistences the Al(OH)_3_ on the worn surface of NiCrAlY coating in NaOH solution in the process of sliding but the following reaction [32]:(8)$$2\mathrm{Al}+2{\mathrm{OH}}^{-}+H_{2}O\;\to 2{{\mathrm{AlO}}_{2}}^{-}+2H_{2}$$

Of course, in addition to the Al dissolution, according to the standard of electrode potentials, Cr element is also dissolved at the anodic cycle and is electrochemically oxidized to Cr_2_O_3_, which is well consistent with the micro-Raman results (Figure 7) [1]. The oxidation reaction process can be illustrated as follows [30]:(9)$$2\mathrm{Cr}+3H_{2}O\;\to {\mathrm{Cr}}_{2}O_{3}+6H^{+}+6e^{-}$$

The above oxidation reactions and metal dissolution explain the smooth surface. Al(OH)_3_ can be evenly distributed on the smooth worn surface and effectively reduce the direct contact of counter ball and sample. At the same time, the frictional shear stress can form the lubricating layer on the worn surface, which can obviously reduce the friction coefficient of coating in HCl corrosive solution [1]. Therefore, the COF of NiCrAlY coating in the HCl corrosive solution is the lowest of 0.24 (Figure 5).

## 4. Conclusions

In this work, the corrosion-wear properties of NiCrAlY coating were studied under deionized water, artificial seawater, 0.1 M HCl solution and 0.1 M NaOH. The main conclusions are given as follows:(1)The NiCrAlY coating is composed of Ni_3_Al, nickel-based solid solution, NiAl and Y_2_O_3_.(2)In deionized water, the NiCrAlY coating with the lowest corrosion current density of 7.865 × 10^−8^ A/cm^2^ is hard to erode. Meanwhile, it presents a lower friction coefficient and the lowest wear rate.(3)In HCl corrosive solution, the coating gives the highest corrosion current density (*i_corr_*) of 3.356 × 10^−6^ A/cm^2^ and a higher wear rate of 6.36 × 10^−6^ mm^3^/Nm.(4)In HCl corrosive solution, the emergence of Al(OH)_3_ on the coating surface could reduce the direct contact between the counter ball and sample effectively, which is conducive to the lowest friction coefficient of 0.24.

## Figures and Tables

**Figure 1 materials-13-01864-f001:**
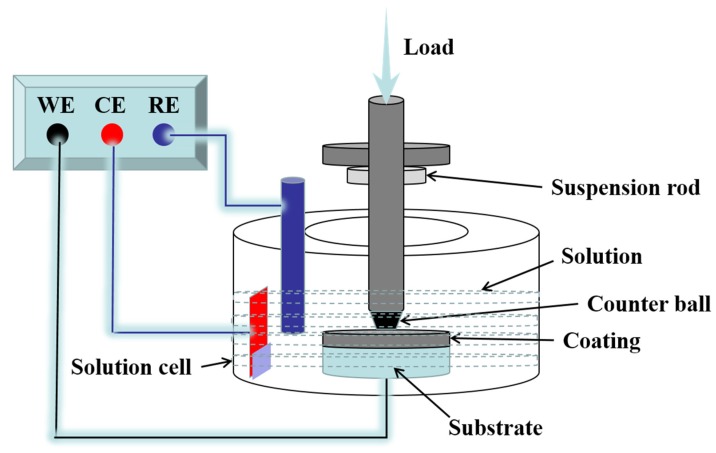
Reciprocating ball-on-disc tribometer schematic diagram.

**Figure 2 materials-13-01864-f002:**
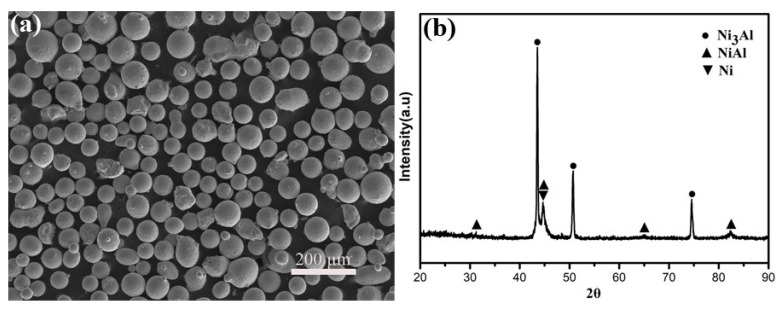
SEM micromorphology (**a**) and XRD pattern (**b**) of NiCrAlY powder.

**Figure 3 materials-13-01864-f003:**
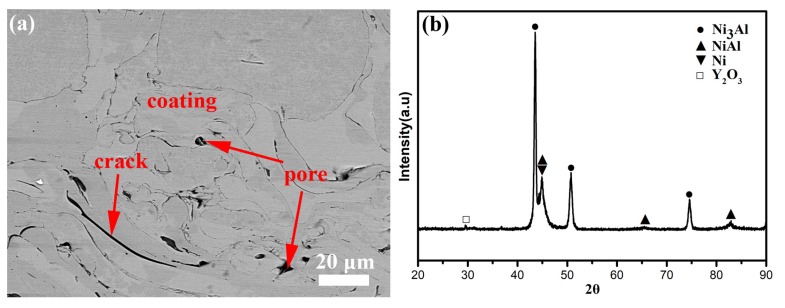
SEM morphology of (**a**) cross-section and (**b**) XRD pattern of NiCrAlY coating.

**Figure 4 materials-13-01864-f004:**
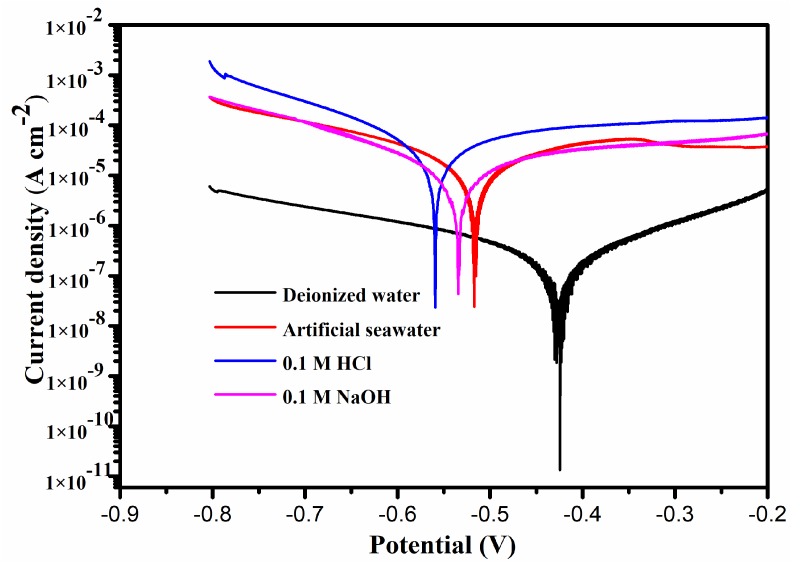
Potentiodynamic polarization curves of NiCrAlY coating sliding conditions in different corrosive solutions.

**Figure 5 materials-13-01864-f005:**
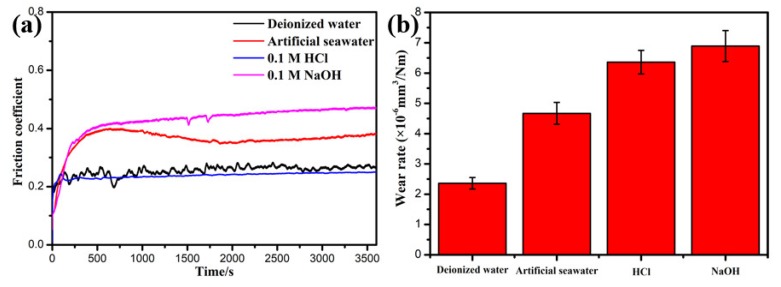
Friction curves (**a**) and wear rate (**b**) of NiCrAlY coating in different corrosive solutions.

**Figure 6 materials-13-01864-f006:**
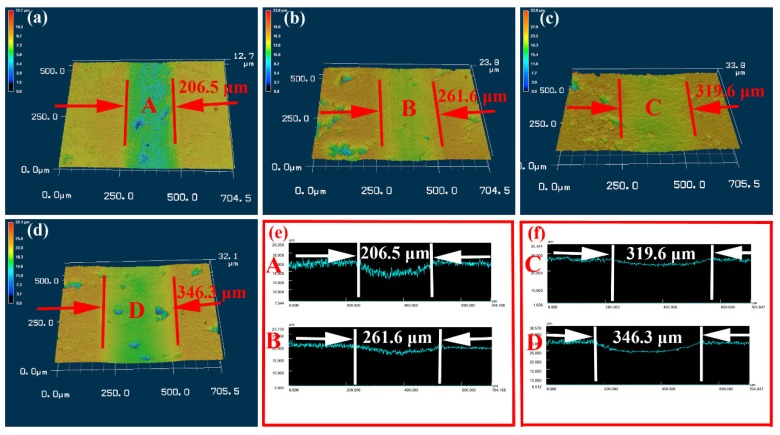
2D and 3D configurations of NiCrAlY coating worn surfaces in different corrosive solutions: (**a**) deionized water, (**b**) artificial seawater, (**c**) HCl solution and (**d**) NaOH solution; (**e**) 2D profiles of A and B regions; (**f**) 2D profiles of C and D regions.

**Figure 7 materials-13-01864-f007:**
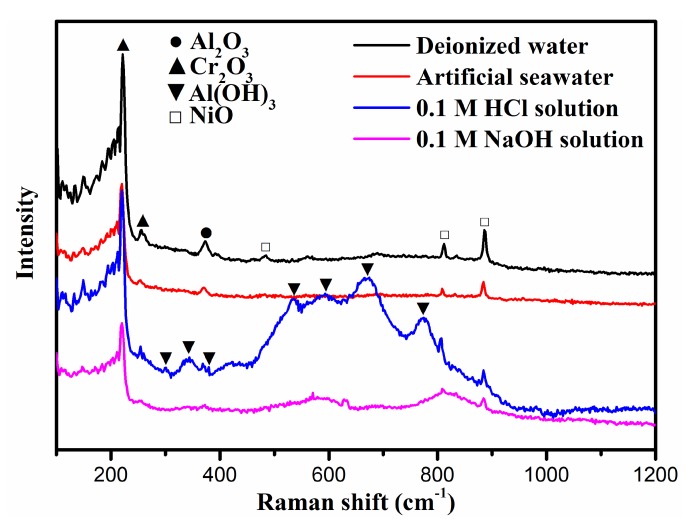
Raman spectra of worn surface of NiCrAlY coating in different corrosive solutions.

**Figure 8 materials-13-01864-f008:**
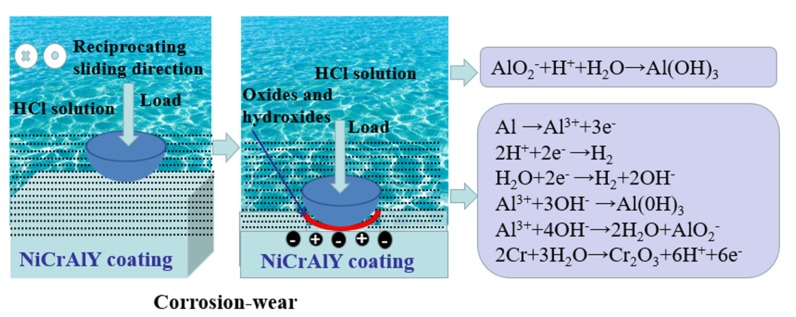
Schematic diagram of corrosion-wear mechanisms of NiCrAlY coating in HCl solution.

**Table 1 materials-13-01864-t001:** Corrosion parameters of NiCrAlY coating from potentiodynamic polarization curves.

Corrosive Solutions	*E_corr_* (V, vs. SCE)	*i_corr_* (A/cm^2^)	*β_a_* (V/dec)	*−β_c_* (V/dec)	*R_p_* (Ω)
Deionized water	−0.428	7.865 × 10^−8^	0.072	0.049	1.610 × 10^5^
Artificial seawater	−0.516	8.986 × 10^−7^	0.043	0.042	1.027 × 10^4^
0.1 M HCl	−0.559	3.356 × 10^−6^	0.036	0.039	2.422 × 10^3^
0.1 M NaOH	−0.535	1.039 × 10^−6^	0.039	0.038	8.044 × 10^3^

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
