# Peer review of "Tribocorrosion Properties of NiCrAlY Coating in Different Corrosive Environments"

_materials, 2020, doi:10.3390/ma13081864_

Round 1

Reviewer 1 Report

Review on the manuscript „Tribocorrosion Properties of NiCrAlY Coating in Different Corrosive Environments”.

The current form of the manuscript does not meet the criterions of quality and requires major revisions.

Some observations are summarized below:

Please carefully proof-read spell check to eliminate grammatical and spelling errors.

I.e. line 65 coating instead coting, Fig. 3 Load instead Loard, etc

Authors should make the manuscript more appealing by highlighting the novelty of the current research. The chemical composition of the powder is not mentioned. Figure 3a does not illustrate a significant caption of the cross section of the coating. An adjustment of the contrast is required, as well as the BSE mode. Maybe the micrograph should be taken at higher magnification. Line 93 what does it mean “dense structural characteristic” What is the porosity level of the coating? The corrosion current densities presented in Table 1 are not in agreement with the polarization curves presented in Figure 4. The corrosion resistance usually is given by the current density. The authors mentioned that “the coating under deionized water shows the lowest corrosion rate” (line133). The polarisation curve of the coating in deionised water reveals the highest current density which means the lowest corrosion resistance. The corrosion mechanism from section 3.4 is doubtful. Please revised it carefully. Conclusions should be rewritten in accordance with the correct electrochemical results mentioned above. The paper is very similar with the first article from the reference list!!!

Author Response

Point 1: Please carefully proof-read spell check to eliminate grammatical and spelling errors. I.e. line 65 coating instead coting, Fig. 1 Load instead Loard, etc

Response 1: We are very sorry for our careless. According to the reviewer’s suggestion, we have checked the whole manuscript carefully and eliminated grammatical and spelling errors. Revised portions are marked in red in the revised manuscript.

Figure 1. Schematic diagram of reciprocating ball-on-disc tribometer.

Point 2: Authors should make the manuscript more appealing by highlighting the novelty of the current research.

Response 2: The reviewer’s comment is of scientifically importance. According to the reviewer’s suggestion, we have made the manuscript more appealing by highlighting the novelty of the current research. Revised portions are marked in red in the revised manuscript.

Point 3: The chemical composition of the powder is not mentioned.

Response 3: According to the review’s suggestion, we have added the chemical composition of the powder in the revised manuscript. Corresponding revisions have been added in line 56 in the revised manuscript.

Point 4: Figure 3a does not illustrate a significant caption of the cross section of the coating. An adjustment of the contrast is required, as well as the BSE mode. Maybe the micrograph should be taken at higher magnification.

Response 4: According to the review’s suggestion, we have provided the BSE mode and higher magnification morphology of the cross section of the coating in the revised manuscript.

Figure 3. SEM morphology of (a) cross-section and (b) XRD pattern of NiCrAlY coating.

Point 5: Line 93 what does it mean “dense structural characteristic” What is the porosity level of the coating?

Response 5: We are very sorry for our careless. Actually, the sentence means that every phase combines well and between any two phases have no evident cracks. Corresponding revisions have been added in line 93 in the revised manuscript.

Point 6: The corrosion current densities presented in Table 1 are not in agreement with the polarization curves presented in Figure 4.

Response 6: We are very sorry for our careless. According to the review’s suggestion, we retest the potentiodynamic polarization curves of NiCrAlY coating sliding condition in different corrosive solutions and calculate the corrosion current density (icorr), corrosion potential (Ecorr), anodic and cathodic Tafel slopes (βa and βc). Corresponding revisions and discussions have been added in the revised manuscript.

Figure 4. Potentiodynamic polarization curves of NiCrAlY coating sliding condition in different corrosive solutions.

Table 1. Corrosion parameters of NiCrAlY coating from potentiodynamic polarization curves.

Corrosive solutions

Ecorr(V, vs.SCE)

icorr (A/cm2)

βa (V/dec)

c (V/dec)

Rp (Ω)

Deionized water

-0.428

7.865×10-8

0.072

0.049

1.610×105

Artificial seawater

-0.516

8.986×10-7

0.043

0.042

1.027×104

0.1 M HCl

-0.559

3.356×10-6

0.036

0.039

2.422×103

0.1 M NaOH

-0.535

1.039×10-6

0.039

0.038

8.044×103

Point 7: The corrosion resistance usually is given by the current density. The authors mentioned that “the coating under deionized water shows the lowest corrosion rate” (line133). The polarisation curve of the coating in deionised water reveals the highest current density which means the lowest corrosion resistance.

Response 7: We are very sorry for our careless. According to the review’s suggestion, we retest the potentiodynamic polarization curves of NiCrAlY coating sliding condition in different corrosive solutions and calculate the corrosion current density (icorr), corrosion potential (Ecorr), anodic and cathodic Tafel slopes (βa and βc). The corrosion current density (icorr) of this coating in deionized water shows the lowest of 7.865×10-8 A/cm2, which means the highest corrosion resistance. Corresponding revisions and discussions have been added from line 108 to line 114 in the revised manuscript.

Point 8: The corrosion mechanism from section 3.4 is doubtful. Please revised it carefully.

Response 8: According to the review’s suggestion, we have revised the corrosion mechanism from section 3.4 in the revised manuscript. Revised portions are marked in red in the revised manuscript.

Point 9: Conclusions should be rewritten in accordance with the correct electrochemical results mentioned above.

Response 9: According to the review’s suggestion, we have rewrote the conclusions in accordance with the correct electrochemical results mentioned above. Corresponding revisions and discussions have been added from line 193 to line 201 in the revised manuscript.

Point 10: The paper is very similar with the first article from the reference list!!!

Response 10: According to the reviewer’s suggestion, we have checked the whole manuscript carefully and made the manuscript more appealing by highlighting the novelty of the current research. Actually, the first article from the reference list is mainly investigated the tribological performance of NiCoCrAlYTa coating under 0.1 M HCl and NaOH solutions. This paper is mainly investigated the corrosion-wear properties of NiCrAlY coating under deionized water, artificial seawater, 0.1 M HCl solution and 0.1 M NaOH. It’s not similar with the first article from the reference list. Revised portions are marked in red in the revised manuscript.

Reviewer 2 Report

Please try to improve the English of the text. An example could be the long sentence in the lines 154-156.

The Introduction must be improved.

The name Raman is not correct in Fig. 7.

In the Abstract many chemical formulas are not written properly (subscripts).

Fig. 8 is really NOT illustrative scientifically in this fashion. Please re-draw it. 

Please check the REAL meaning of the English words, as e.g. is there such a word as Surprisily ? (In line 126.)

Author Response

Point 1: Please try to improve the English of the text. An example could be the long sentence in the lines 154-156.

Response 1: According to the reviewer’s suggestion, we have improved the English of the whole manuscript carefully and eliminated grammatical and spelling errors. Revised portions are marked in red in the revised manuscript.

Point 2: The Introduction must be improved.

Response 2: According to the reviewer’s suggestion, we have improved the Introduction in the revised manuscript. Revised portions are marked in red in the revised manuscript.

Point 3: The name Raman is not correct in Fig. 7.

Response 3: We are very sorry for our careless. We have revised it in the revised manuscript.

Figure 7. Raman spectra of worn surface of NiCrAlY coating in different corrosive solutions.

Point 4: In the Abstract many chemical formulas are not written properly (subscripts).

Response 4: We are very sorry for our careless. We have revised the chemical formulas in the Abstract in the revised manuscript. Revised portions are marked in red in the revised manuscript.

Point 5: Fig. 8 is really NOT illustrative scientifically in this fashion. Please re-draw it. 

Response 5: According to the reviewer’s suggestion, we have re-drawn the Fig. 8 in the revised manuscript.

Figure 8. Schematic diagram of corrosion-wear mechanisms of NiCrAlY coating in HCl solution.

Point 6: Please check the REAL meaning of the English words, as e.g. is there such a word as Surprisily ? (In line 126.)

Response 6: We are very sorry for our careless. We have checked the whole manuscript carefully and eliminated grammatical and spelling errors. Surprisingly is the exactly word I want to use. Revised portions are marked in red in the revised manuscript.

Reviewer 3 Report

This seems to be an interesting approach, but unfortunately the paper lacks scientific rigor. Moreover, misleading use of English contributes for further misunderstanding.

English issue examples:

“and had the serious corrosion”

“This research is hoped”

“hold brilliant”

“Current researches basically incline to explore the oxidation resistance at high temperature, corrosion, wear resistance , mechanical and tribological performance of the coatings”

Fig. 1: loard

“because its rate is often scale with the corrosion rate”

“the lowest corrosion rate and may be very hard to corrod”

“Aluminate anions cannot maintain unstable in HCl corrosive solution »

“it don’t formate the”

“The coating in deionized water has the highest corrosion potential means that the coating is the most hard to be corroded”

This is not true. The corrosion potential only indicates in which potential corrosion will naturally occur when coupled to the cathodic process in question. The current densities are the final indication on the corrosion resistance.

How was the wear rate was calculated? This is essential information. Effect of corrosion products formation (volume increase) should be considered besides wear.

The reduction reactions (Eq. 3 and 4) are not universal and should be specified for each medium.

What is the pH for each media?

What is the % of Al in the coating? This is fundamental to confirm hypotheses based on Al-based corrosion products.

What is the % of Y? Y is a rare-earth element that, such as cerium, can work as corrosion inhibitor. This could positively affect the corrosion behaviour of your system.

8: Al2(OH)3 is not stable in acidic media!

Author Response

Point 1: This seems to be an interesting approach, but unfortunately the paper lacks scientific rigor. Moreover, misleading use of English contributes for further misunderstanding.

English issue examples:

“and had the serious corrosion”

“This research is hoped”

“hold brilliant”

“Current researches basically incline to explore the oxidation resistance at high temperature, corrosion, wear resistance , mechanical and tribological performance of the coatings”

Fig. 1: loard

“because its rate is often scale with the corrosion rate”

“the lowest corrosion rate and may be very hard to corrod”

“Aluminate anions cannot maintain unstable in HCl corrosive solution”

“it don’t formate the”

Response 1: We are very sorry for our careless. According to the reviewer’s suggestion, we have improved the English of the whole manuscript carefully and eliminated grammatical and spelling errors. Revised portions are marked in red in the revised manuscript.

Point 2: “The coating in deionized water has the highest corrosion potential means that the coating is the most hard to be corroded”

This is not true. The corrosion potential only indicates in which potential corrosion will naturally occur when coupled to the cathodic process in question. The current densities are the final indication on the corrosion resistance.

Response 2: We are very sorry for our careless. According to the review’s suggestion, we retest the potentiodynamic polarization curves of NiCrAlY coating sliding condition in different corrosive solutions and calculate the corrosion current density (icorr), corrosion potential (Ecorr), anodic and cathodic Tafel slopes (βa and βc). The coating in deionized water has the lowest corrosion current density of 7.865×10-8 A/cm2 means that the coating is hard to corrode. Corresponding revisions have been added in line 109 in the revised manuscript.

Point 3: How was the wear rate was calculated? This is essential information. Effect of corrosion products formation (volume increase) should be considered besides wear.

Response 3: The wear rate was calculated by W = V/LF, where W was the wear rate in mm3/N.m, V was the wear volume loss in mm3, L was the sliding distance in m and F was the load in N. Corresponding revisions have been added from line 78 to line 80 in the revised manuscript.

Point 4: The reduction reactions (Eq. 3 and 4) are not universal and should be specified for each medium.

Response 4: Yes. Actually, the reduction reactions of Eq. 3 and 4 are the electrochemical reactions of NiCrAlY coating in HCl solution.

Point 5: What is the pH for each media?

Response 5: The pH of deionized water is 7. The pH of artificial seawater is 8.2. The pH of NaOH solution is 13. The pH of HCl solution is 1. Corresponding revisions have been added in line 70 in the revised manuscript.

Point 6: What is the % of Al in the coating? This is fundamental to confirm hypotheses based on Al-based corrosion products.

Response 6: The content of Al (wt. %) in the NiCrAlY powder is 10. Corresponding revisions have been added in line 56 in the revised manuscript.

Point 7: What is the % of Y? Y is a rare-earth element that, such as cerium, can work as corrosion inhibitor. This could positively affect the corrosion behaviour of your system.

Response 7: The content of Y (wt. %) in the NiCrAlY powder is 1.0. Corresponding revisions have been added in line 56 in the revised manuscript.

Reviewer 4 Report

There would be even more interesting and comprehensive to perform also dry sliding test at the "same" conditions.

However, the idea is new and interesting and the results look to be valid.

Author Response

Point 1: There would be even more interesting and comprehensive to perform also dry sliding test at the "same" conditions.

However, the idea is new and interesting and the results look to be valid.

Response 1: The reviewer’s comments are of scientifically importance. Actually, this paper is mainly investigated the corrosion-wear properties of NiCrAlY coating under deionized water, artificial seawater, 0.1 M HCl solution and 0.1 M NaOH. This research would provide valuable guidance to the application of NiCrAlY coating as tribomaterials in corrosive environments. We will investigate the tribological performance of NiCrAlY coating in the condition of dry sliding test in the future.

Round 2

Reviewer 1 Report

the paper can be published in  the presented form

Author Response

Point 1: The paper can be published in  the presented form.

Response 1: Thank you very much.

Reviewer 3 Report

Unfortunately, this paper has unacceptable scientific flaws. It must therefore me rejected.

For example:

It confuses corrosion with erosion, it adresses the corrosion resistace of materials that cannot corrode.

The corrosion mechanims presenting from Eqs 2 to 9 and Fig 8 lack experimental proof and have inconsistensies.

Author Response

Point 1: It confuses corrosion with erosion, it adresses the corrosion resistace of materials that cannot corrode.

Response 1: This coating is used in the marine atmosphere, pulping and mining, suffer the collaborative destruction of corrosion and wear, which should be required the excellent corrosion resistance property. This paper mainly investigates the tribocorrosion mechanism of NiCrAlY coating in different corrosive environments. In recent years, some researchers investigate the tribocorrosion properties of Ni-based composite coatings. The results indicate that the Ni-based composite coatings have the excellent tribocorrosion properties in corrosion environment.

Point 2: The corrosion mechanims presenting from Eqs 2 to 9 and Fig 8 lack experimental proof and have inconsistensies.

Response 2: J. Chen et al. investigated the tribocorrosion properties of NiCoCrAlYTa coating in different corrosion environments. The corrosion mechanism of NiCrAlY coating is very similar with that NiCoCrAlYTa coating in HCl solution. The Raman results indicate that the corrosion product of Al(OH)3 is formed in HCl solution. The local corrosion is occurred on the surface of NiCrAlY coating (Fig. 6).

Round 3

Reviewer 3 Report

A few English corrections are presented in the attached pdf, together with a few corrections of misconception of concepts.

After these reviews, the article can be published.

Author Response

Point 1: A few English corrections are presented in the attached pdf, together with a few corrections of misconception of concepts.

Response 1: According the reviewer’s suggestion, we have revised the English corrections in the revised manuscript.